# Nutritional Balance and Genetic Diversity of *Coffea canephora* Genotypes

**DOI:** 10.3390/plants12071451

**Published:** 2023-03-26

**Authors:** Maria Juliete Lucindo Rodrigues, Cleidson Alves da Silva, Heder Braun, Fábio Luiz Partelli

**Affiliations:** 1Graduate Program in Tropical Agriculture, Federal University of Espírito Santo, São Mateus 29932-540, ES, Brazil; 2Programa de Pós-Graduação em Fitotecnia, Federal University Lavras, Lavras 37200-000, MG, Brazil; 3Graduate Program in Agroecology, State University of Maranhão, São Luís 65055-310, MA, Brazil; 4Department of Agrarian and Biological Sciences, Federal University of Espírito Santo, Vitória 29500-000, ES, Brazil

**Keywords:** nutrient export, genetic diversity, fertilization efficiency, conilon coffee

## Abstract

Our objectives were to study characteristics of the fruit (weight, percentage of husk/grain), to determine the concentration and accumulation of nutrients in the fruits, grain and husk, and to verify the existence of genetic diversity in *Coffea canephora* genotypes. The experiment was conducted with 20 genotypes in a four-year-old plantation, in a randomized block design with four replications and five plants per plot. The fruits were oven-dried, depulped (husk separated from the grain) and sent to a laboratory for nutritional analysis. Macronutrients N and K were the most accumulated/exported in fruits, respectively. In addition, the different genotype control cycles influenced the accumulation of nutrients in the fruits. There was genetic diversity among the 20 *C. canephora* genotypes, studied for the characteristics of concentration and percentage of grain/straw nutrients in the fruit. Genotypes 2, 8 and 13 were the ones with the greatest genetic distance, consequently they are the most dissimilar when compared to the other genotypes. Genotypes 8 and 1 stand out for having a higher proportion of fruit weight in relation to grains. Therefore, they are the genotypes that need a smaller amount of fruit to produce 1000 kg of ground coffee.

## 1. Introduction

In the agricultural sector of the world economy, Brazil is the largest producer and exporter of coffee [1]. The best-known and economically most relevant coffee species in the world are *Coffea arabica* L *(arabica*) and *C. canephora* Pierre ex A.Froehner (conilon and/or robusta) [2]. Conilon coffee is grown at low altitudes in several Brazilian states, mainly in Espírito Santo, Bahia and Rondônia [3].

As a crop with a high yield potential, which accumulates large nutrient amounts in the vegetative and reproductive tissues, coffee requires high levels of nutrient supply [4]. Consequently, the nutritional management has to ensure a balance between the amount of supplied nutrients and those removed in the harvested fruits.

The reproductive phase consists of five well-defined stages until the fruits are harvested. Fruit set begins soon after flowering, characterized by the stages: initial fruit growth, rapid expansion, suspended growth, grain-filling and maturation [5].

Temporarily, the flowering and fruiting periods act as the main temporary accumulators of nutrients in the plant [6]. When fruit development begins, these nutrients are slowly translocated to the developing organs, followed by a period of greater nutrient concentration and accumulation that coincides with the rainy season, of fruit set and maturation and finally, nutrient uptake stabilizes again as the cycle come to an end [7].

The main nutrients required by coffee are accumulated in the plant tissues in the following order: nitrogen (N) > potassium (K) > calcium (Ca) > phosphorus (P) = magnesium (Mg) = sulfur (S) e iron (Fe) > boron (B) > manganese (Mn) > copper (Cu) > zinc (Zn) [8]. Although the required amounts differ, mainly according to the age and productivity of each genotype, these nutrients are essential for coffee growth, development and production [9].

The accumulation curves in *C. canephora* differ according to the plant genotypes [7]. The *C. canephora* genotypes are grouped according to their maturation cycle (early, medium, late), distinguishable by the distinct numbers of days to complete the cycle from flowering to fruit maturation. This makes an improved exploitation of the applied nutrients possible, and consequently, gains in productivity [10].

Based on the data of studies on nutrient concentrations in conilon fruit, the nutritional management can be adjusted and the genotypes with more homogeneous and more relevant characteristics can be selected. In addition, analysis provides essential answers about calculating the nutrients responsible for crop productivity [5]. Therefore, this study focused on the search for genetic variability in the nutrient concentration and accumulation in the grain and husk of *C. canephora* genotypes, and to estimate the level of nutrient withdrawal in the harvested fruits of these genotypes.

## 2. Results

### 2.1. Genetic Parameters and Nutrient Concentrations in Grain and Husk

All nutrients in the grain and husk were affected by their genotypes, except for the N and P concentrations in the grain and the S, Zn and B concentrations in the husk, which did not differ among the genotypes (Table 1).

In a comparison of the fruit components, the grain concentrations were the highest for the macronutrients N (29.28), K (17.01) and P (1.79), while in the husk, the concentrations were highest for K (24.99), N (16.67) and Ca (4.61). For the micronutrients, the concentrations of Mn (19.40), Fe (19.12) and B (10.37) were the highest in grain and husk, and for macronutrients, the concentrations of Fe (22.68), B (21.02) and Mn (16.71) in grain and husk. In addition, the macronutrients and micronutrients can be classified, respectively, in descending order for the concentration in grain: N > K > P > Mg > Ca = S and Mn > Fe > B > Cu > Zn, and in husk: K > N > Ca > S > P > Mg and Fe > B > Mn > Zn > Cu.

In the mean, the fruits of the genotypes contained a higher percentage of grain (56.95%) than of husk (43.05%). Therefore, for each ton of yield, the weight of the green coffee is higher than that of the husk after processing.

In general, the coefficient of experimental variation (CVe) was lower than 20% for macronutrient and micronutrient concentrations, except for Fe (28.92%) and Zn in the grains (23.17%) and the S concentration in the husk (20.67%) (Table 1).

The CVg that quantifies the influence of the genetic components on each characteristic ranged from 2.98% (N in grain) to 40.21% (Fe in grain). Values of CVg lower than 10% were observed for the N, P, K and S concentrations in grain and for the N, P, S, Zn and B concentrations in the husk (Table 1).

The N, P and K grain concentrations had a heritability index (H^2^) below 50%, while those of Ca, Mg, Cu, Fe and Mn had a H^2^ of more than 80% (Table 1). The index of the B husk concentration was the lowest (25.9%), while the H^2^ of the husk concentrations of N, P, K, Ca, Mg, Cu and Mn exceeded 80% (Table 1).

For most nutrients, variability was identified among the 20 genotypes for grain concentration, except for N and P (Table 2). In general, five groups were formed for grain nutrient concentrations, depending on the characteristic.

The genotypes were divided into two dissimilar groups for grain concentrations of K, Ca, S, Zn and B and separated into three groups for Mg, Fe and Mn. In genotype 2, inversely to the pattern of Cu, the highest means were found for the grain concentration of the three nutrients: Mg (2.23), Fe (42.33) and Mn (23.43).

For the grain concentration of the micronutrient Cu, a greater variability among genotypes was observed, resulting in five dissimilar groups. Genotypes 3 and 13 were grouped similarly in groups with the highest means (12.17 and 12.50), and genotypes 2, 12 and 19 were assigned to the group with lowest means (6.87, 7.03 and 6.27) for grain concentration. Genotype 2 appeared most frequently in the group of highest means and was assigned to the group with the highest means of six nutrients (K, Ca, Mg, Fe, Mn and Zn).

Variability in husk concentration among the 20 genotypes was identified for most nutrients, except for S, Zn and B (Table 3). In general, groups were formed for husk nutrient concentrations.

Six groups of mean concentrations were formed for the nutrient Mg. Only genotypes 4 and 8 were assigned to the groups with highest (1.23) and lowest mean (0.53), respectively, for Mg concentration in the husk.

The micronutrients Cu and Mn were clustered into five groups. For Mn, low means were observed for the genotypes 7, 8, 11, 15, 17 and 19 (between 8.90 and 11.93) and for Cu, only genotype 2 had a very low mean (4.07) and genotype 16 the highest mean concentration (8.63).

The macronutrients N, P, K and Ca formed three groups. Nitrogen and K were the only nutrients with a higher number of genotypes grouped with highest means (from 16.80 to 29.40), comprising 10 and 16 genotypes, respectively.

### 2.2. Nutrient Accumulation in Fruits

Variability in all nutrients accumulated in the fruits was identified among the 20 genotypes (Table 4). In general, groups were formed for fruit nutrient concentrations.

The genotypes were divided into 10 dissimilar groups for the Mg and Mn accumulation, representing the variables with the highest variation. For Mg, only genotypes 2 and 13 had high means (2.83 and 2.78) and for Mn, only genotype 2 (45.89). Nutrients N and B were divided into six groups. Only genotype 13 was classified in the group of the highest means (43.37 and 33.94, respectively).

The same genotypes that were clustered in the group of highest means for certain nutrients also appeared in the group of lowest means for others. Genotype 13 appeared most frequently in the group of highest means, and was clustered in the best group for seven nutrients (N, P, K, Mg, Cu, Zn and B). On the other hand, 70% of the genotypes (1, 3, 5, 7, 8, 9, 10, 11, 12, 14, 15, 16, 17 and 20) were not assigned to the highest mean group for any of the nutrients evaluated for fruit nutrient accumulation.

### 2.3. Characteristics of Grain, Husk and Fruit

The data of the variables analyzed differed significantly between the genotypes. The relationship between the grain and the husk discriminated the 20 genotypes in five groups (Figure 1).

The first group, with the highest grain percentages, consisted of four genotypes (7, 8, 16 and 17), the second of six (20, 3, 12, 11, 5 and 10), the third of five (1, 2, 19, 9 and 18), the fourth of four genotypes (4, 15, 14 and 6) and the fifth group of only one (genotype 13).

In group I, genotypes have a higher grain yield in relation to the husk (from 59.53 to 61.11%), indicating excellent genotypes for use in breeding. Genotype 13, in the fifth group, had a lower relationship between the husk and grain (51.84%); in this case, the genotype is unattractive for production, because grain and husk production are almost equal.

The variables grain weight and fruit weight varied significantly among the 20 genotypes. The Scott Knott test detected variability among the genotypes for the characteristics of coffee fruit (Figure 2). The variable fruit weight divided the 20 genotypes into five groups and grain weight into seven groups.

Genotypes 8 and 1 constitute the first group, corresponding to the highest ratio of fruit weight to grain weight. The second group also comprises two genotypes (19 and 15) for fruit weight and only genotype 15 for grain weight. The third group for fruit weight was formed by the highest number of genotypes (9, 3, 12, 5, 16, 20 and 11) and for grain weight by the genotypes 19, 9, 12 and 5. The fourth group consisted of genotypes 10, 2, 17, 14 and 18 for fruit weight, and for grain weight, of genotypes 3, 16, 20 and 11. Finally, the fifth group for fruit weight was formed by four genotypes (13, 7, 4 and 6) and for grain weight by genotypes 10, 2, 17, 18 and 7. Genotype 14 and genotypes 13, 4 and 6 correspond to the groups VI and VII, respectively, for grain weight.

### 2.4. Cluster Analysis

Cluster analysis performed by the hierarchical method UPGMA (dendrogram formation) and the Tocher method using the Euclidean distance as dissimilarity measure revealed wide genetic variability among genotypes, forming six and five groups, respectively (Figure 3), at a 70% dissimilarity threshold between genotypes.

Using the UPGMA method, the 20 genotypes were clustered as follows: group I comprised genotypes 4, 10 and 16; group II the genotypes 18, 20, 7 and 3; and group III was the most representative, as it included 10 (50%) of the genotypes (15, 19, 11, 9, 6, 17, 1, 12, 5 and 14). The other three groups (groups IV, V and VI) contained one genotype each (8, 13 and 2, respectively) (Figure 3).

However, using the Tocher method, the number of groups was reduced to five, although genotypes 8, 13 and 2 were maintained in different individual groups. The characteristics of these three genotypes differed from those of the others and were therefore assigned to three individual groups.

For genotype 2, the highest means were identified for more than 60% of the nutrients studied, with regard to grain concentration: K (17.97), Ca (1.90), Mg (2.23), Fe (42.33), Mn (23.43) and Zn (10.50), and for husk concentration: N (18.90) and Mn (35.03). For this genotype, micronutrient concentrations were the highest for Mn in both evaluated characteristics.

Genotype 8 stood out with the highest levels of K (17.83) and Zn (10.40) in grain nutrient concentrations and for the husk concentration of the micronutrient Fe (32.23).

For genotype 13, more than 50% of the nutrients were grouped as the highest means for the trait grain concentration. The concentrations of the primary macronutrients N (18.43), P (1.27) and K (25.50) were highest in the husk. Phosphorus was the only nutrient in the group of the highest means that appeared only for genotype 13, for the traits of husk or grain concentration.

For the grain to husk percentage, genotype 8 had a lower ratio, indicating a greater grain than husk production of this genotype. The grain/husk ratio of genotype 13 was the highest, indicating a lower grain yield than of the other genotypes.

The most similar genotypes were 15 and 19, which were grouped in the large group with half of the genotypes. For the characteristics evaluated in this study, these two genotypes were close to each other and had lower genetic diversity. Dissimilarity was the greatest between genotypes 4 and 2, and for the characteristics evaluated in the study, they were considered the most distinct.

To determine the relative contribution of the variables to genetic diversity among the 20 genotypes in relation to grain and husk nutrient concentrations, the Singh method [11] was used, resulting in values from 0.003 to 43.67% (Figure 4).

The micronutrients Fe in the coffee grain (43.67%) and Mn in the husk (25.37%) contributed most to genetic diversity; together they accounted for 69.04% of the variability. The Fe grain and Mn husk concentrations were also the variables with highest CVg, which reinforces the result of the contribution of these nutrients to genetic diversity. The other nutrients contributed with less than 7%.

## 3. Discussion

### 3.1. Genetic Parameters

The significant results confirmed the existence of genetic variability among the genotypes, i.e., these characteristics contribute to breeding research for possible indications of superior genotypes [12,13].

For almost all nutrients and organs studied, the experimental coefficient of variation (CVe) was always <20% (Table 1). This shows the low environmental influence and high precision in the experiment, and the value is in the range considered acceptable for experiments with perennial crops, such as coffee [14]. However, for the micronutrients Fe (28.92%), Zn (23.17%) in grains and S (20.67%) in the husk, values above 20% were observed.

Evaluation studies conducted by [4,8,13,15] with nutrient concentrations in different organs (leaf, flowers, grains, husk, fruits) of conilon coffee also reported values higher than 20% for some nutrients. Factors such as size of the experiment, maturation cycle and genotype response to biotic and abiotic factors may have contributed to the high indices [14].

The genetic coefficient of variation (CVg) determines the influence of genetic components for each characteristic, i.e., the higher the coefficient, the greater the genetic influence. According to [16], CVg values above 7% are considered high. However, in this study, the nutrient concentrations were lower in the grain for N, P and K and in the husk for P, Zn and B, i.e., genetic variability was below the desirable criterion for breeding research with these elements [15,17] (Table 1).

The nutrients Ca, Mg, Cu and Mn exceeded 80% in the grain and the husk. For heritability (H^2^), indices above 80% are considered highly relevant for breeding, since the trait expresses the confidence degree of the phenotypic value, which is a favorable genetic indicator for selection [4,18]. In an evaluation of nutrient concentrations in flowers, leaves, grains and husks in Robustas Amazônicos, [4] also found values above 80% for some nutrients.

### 3.2. Nutrient Accumulation in Fruit and Concentrations in Grain and Husk

In each phase, nutrients are reallocated according to the plant’s requirements and preference for physiological and metabolic functions [6,7]. The allocated amount varies between genotypes, management systems (irrigated and rainfed) and altitude [19,20].

Significant differences in the grain and husk nutrient concentrations between *C. canephora* genotypes cv. Robusta were also reported by [4] and in leaf concentrations by [15]. The variation in the different organs may be associated with the genotype-specific requirements throughout the cycle, nutrient uptake, mobility in the conductive vessels and distribution of the root system [21].

Nitrogen (N) is a primary macronutrient found in the highest amounts in the leaves and fruits of coffee plants [8]. Studies also emphasize that at harvest, N is the most extracted nutrient in *C. canephora* fruits [5], and is the second most extracted in *C. arabica* fruits, where potassium is the first [22].

Among micronutrients, Fe is considered to be the nutrient with the highest accumulation in fruits (Table 4), which confirms the findings of [20,23] for *C. canephora* fruits at full maturity. According to [9], Fe was also found to be the most common micronutrient in 72-month-old robusta coffee trees. In an evaluation of 6-month-old conilon coffee seedlings, [24] also reported higher levels of Fe and [25] found higher accumulations of Fe in fruits for all genotypes in a study on conilon genotypes with different maturation cycles.

The dried fruits consist of the husk/skin, junction of the epicarp, mesocarp and endocarp and of grains with a denser structure (endosperm), usually separated in two grain halves [26]. The grains are marketed, and the husk is considered only a waste product generated by fruit processing. However, studies by [5,27] emphasize the presence of high nutrient rates contained in this residue, e.g., potassium and nitrogen.

In view of the high fruit production and since at harvest high amounts of nutrients are extracted from the plantation, coffee crops require the replacement of these nutrients. Studies such as this are therefore extremely important, because they can be used to calculate nutrient exports and adjust the subsequent fertilizations according to the expected crop productivity. It is also possible to quantify the nutrients that can be replaced contained in the coffee husk, an excellent low-cost option and source of some nutrients.

According to [6,28], nutrient accumulation curves in *C. canephora* genotypes cvs. conilon/robusta have the same pattern and differ only in relation to the maturation cycles (early, medium or late). The efficiency in nutrient uptake may vary among genotypes, as stated for arabica coffee cultivars [29].

In an analysis of macronutrient accumulation in conilon coffee, [9] observed that the nutrient demands among the species of the genus *Coffea* differ. In addition, each genotype has an accumulation pattern in relation to plant age [7]. In general, the N, Ca and K demands of the *C. canephora* species are high.

Nutrient accumulation studies are relevant to make more consistent decisions regarding fertilizer use, in view of the different demand of genotypes to complete their cycle (early, medium and late), and due to the higher nutrient requirement in specific phases. The supply of nutrients should be adjusted to the response of the genotype and the possibility of the splitting of fertilization.

### 3.3. Characteristics of Grain, Husk and Fruit

The weight and grain percentages in the fruits are correlated with yield. According to [30], genotypes with the lowest grain percentages produced the lowest yields and the highest ratio of mature fruit weight and mature fruit volume.

In agreement with our observation of genotypes with higher and lower grain yield per fruit, [30] reported a mean of 62.02% for the highest grain yield per fruit and 51.08% for the lowest yield in a study with 43 conilon coffee genotypes. In that study, the authors observed great variation between genotypes with regard to the volume required to produce a bag of 60 kg and 1000 kg green coffee.

According to [5,31], of the total dry weight of the conilon fruit, the grains correspond to 65% and the husk to 35%, i.e., a ratio of approximately 2:1. For arabica coffee, ref. [32] observed a variation of 43.7% to 55.6% in grain percentage at the fruit processing.

According to [33], from the commercial point of view and for genetic studies, genotypes are preferred which, among other characteristics, have high grain percentages and fruit weight, since a smaller amount of grains is required to produce a 60 kg bag.

### 3.4. Dissimilarity between Genotypes and Relative Contribution of Nutrient Concentration

The study of multivariate analyses is essential for planning strategies and advances in breeding. By the formation of dissimilarity groups, the genetic diversity among *C. canephora* genotypes can be visualized [34,35]. The UPGMA cluster method is commonly used to study genetic diversity in *C. canephora* [8,15] and also in *C. arábica* [35].

To increase the reliability of the dissimilarity among the grouped genotypes, the Tocher method has also been used. In the studies conducted by [4,36,37,38,39], the similarity and consistency of the methods in the formation of groups of *C. canehora* genotypes has been confirmed.

The two clustering methods had very similar results of group composition, mainly because they organized the same genotypes into individual groups, reinforcing the high dissimilarity degree of these in relation to the others [40,41].

These results corroborate those of [4] who studied nutrient concentrations in the different organs of 16 Robusta coffee genotypes (cutoff point 82%). Six groups were formed, of which three comprised only one genotype. In a study on the morpho-agronomic characteristics in conilon coffee, [42] also reported the formation of groups with only one genotype. In an evaluation of the root system distribution of six conilon coffee genotypes (with a cutoff value of 92.23%), [43] observed the formation of groups with only one genotype.

The results of this study diverged from the findings of [15], who highlighted the N concentration in the plant tissues of *C. canephora* as the nutrient that least contributed to genetic diversity, using Singh’s method [11]. The results with regard to the contribution to genetic divergence clearly showed which characteristics were the most relevant in this respect and those that can be excluded for having the lowest value [32].

## 4. Materials and Methods

### 4.1. Location and Genotypes

The experiment was conducted on a private property in the district of Vila Valério (18°57′48″ S, 40°20′08″ W, 110 m asl) in the north of the Espírito Santo, Brazil. In this region, the maximum/minimum temperatures are 29 and 18 °C, respectively. The climate is tropical, type Aw, with characteristically dry winters and rainy summers, according to Köppen’s classification [44]. The soil was classified as Latossolo Vermelho Amarelo distrófico [45] and the soil chemical and physical properties are listed in Table 5.

The coffee trees were four years old and had been harvested three consecutive times. The experiment used 20 conilon genotypes, planted from cutting-propagated seedlings in 2017. The seedlings were planted at a row spacing of 3 m and a plant spacing of 1.20 m.

The cultural treatments were applied according to the crop needs to optimize the phytosanitary and nutritional crop management, and the water demand during dry periods was met by micro-sprinkler irrigation with emitters spaced at 12 × 12 m. (Figure 5).

The genotypes 1 to 19 were visually selected on a commercial plantation of coffee-rubber intercropping after a long regional drought period (end of 2014 to 2015). These genotypes stood out during the dry spell, with exceptionally good yield stability. Genotype 20 corresponds to “genotype 02” and belongs to the cultivar Emcapa 8111 [46].

### 4.2. Fruit Sampling and Evaluations

The treatments consisted of the 20 *C. canephora* genotypes. The experiment was conducted in randomized blocks with four replications. There were five plants per experimental unit, i.e., a total of 400 plants were used. The fruits were collected manually at full maturity. The fruits of the genotypes with early (11,14,16,18) and medium maturation cycles (1, 5, 7, 8, 9, 10, 15, 20) were harvested in May, while the fruits of the late maturation genotypes (2, 3, 4, 6, 12, 13, 17, 19) were harvested in June 2021.

For each genotype, the ripe fruits were manually harvested from five plants per plot, of which a sample of approximately 2000 g was taken and sent to the laboratory. The fruits were placed on plastic trays and stored in a forced ventilation oven at 50 °C to constant mass.

After drying, the material was prepared to determine the relationship of the grain to the husk. For each genotype, the fruits were counted and weighed on a precision scale. Four subsamples of 30 fruits were separated, i.e., a total of 120 fruits per genotypes, and the fruits were processed manually (separation of the grain from the husk).

For chemical analysis, grain and husk samples wrapped in Kraft paper were adequately identified for each genotype and sent to the laboratory. After threshing, the husk and the grain were ground in a Willey mill to determine the concentrations of nitrogen (N), phosphorus (P), potassium (K), calcium (Ca), magnesium (Mg), sulfur (S), iron (Fe), manganese (Mn), copper (Cu), zinc (Zn) and boron (B) by the methodology of [47]. All analyses were performed in triplicate.

Fruit nutrient accumulation was evaluated as the sum of the components’ nutrient accumulation in grain (kg) + nutrient accumulation in husk (kg) [48]. The weight of one dry fruit or grain per genotype was calculated by the following formula: number of fruit or grain/dry fruit or grain weight (g). The percentage of the husk and the grain was calculated as dry fruit or husk weight (g)/dry fruit weight × 100. For all calculations, the moisture content of the fruits (grain + husk) was adjusted to 12%, i.e., the moisture at which the grains are marketed.

### 4.3. Statistical Data Analysis

The data sets were subjected to analysis of variance (ANOVA). The Scott–Knott test (*p* ≤ 0.05) was used to compare the mean nutrient concentrations and accumulated nutrient contents in the grain and husk affected by genotypes. For each evaluated characteristic, the coefficient of the experimental variation (CVe), coefficient of genetic variation (CVg) and heritability (H²) were also estimated. The parameters were estimated by the following formulas:CVe (%) = (σ_e_/M) × 100(1) where σ_e_ = standard deviation of the experimental residue; M = experimental mean.
CVg (%) = (σ_g_/M) × 100(2) where σ_g_ = genetic standard deviation; M = experimental mean.
H^2^ (%) = (σ^2^ _g_/σ^2^
_F_) × 100(3) where σ^2^
_g_ = genotypic variance component; σ^2^
_F_ = component of phenotypic variance.

For the study of genetic diversity, the Euclidean distance matrix was used as a dissimilarity measure. Subsequently, the genotypes were grouped by the Tocher optimization method and the hierarchical unweighted pair group method using arithmetic means (UPGMA). The relative importance of the traits for genetic diversity was evaluated as proposed by Singh [11]. Spearman’s correlation analyses were used to detect possible associations between two variables evaluated. All analyses were performed using software Genes [49].

## 5. Conclusions

The macronutrients N and K were the most accumulated/exported in the fruits, respectively. Therefore, it should be added to larger meals compared to other nutrients. In addition, the different genotype control cycles influenced the accumulation of the nutrients in the fruits.

There was genetic diversity among the 20 *C. canephora* genotypes studied for the characteristics of the concentration and percentage of grain/straw nutrients in the fruit. Genotypes 2, 8 and 13 were the ones with the greatest genetic distance, and consequently, they are the most dissimilar when compared to the other genotypes.

Genotypes 8 and 1 stand out for having a higher proportion of fruit weight in relation to grains. Therefore, they are genotypes that need a smaller amount of fruit to produce 1000 kg of ground coffee.

## Figures and Tables

**Figure 1 plants-12-01451-f001:**
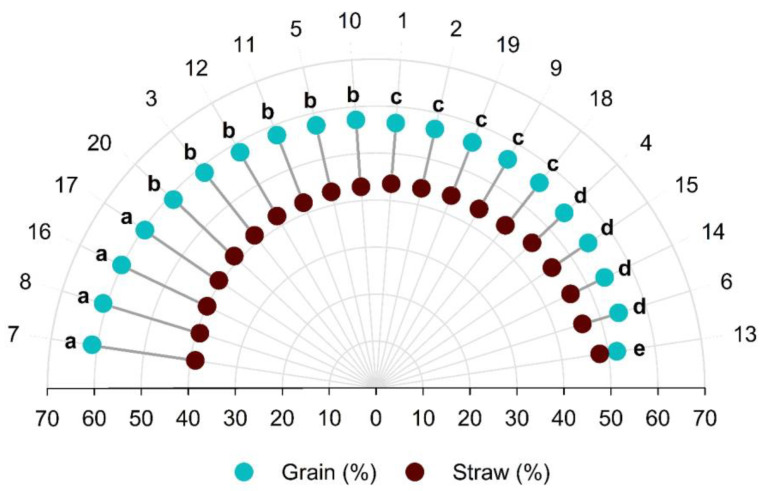
Grain percentage and husk in the fruits of 20 *C. canephora* genotypes dried at 12% moisture. Means followed by the same letter between genotypes did not differ statistically by the Scott–Knott test at 5% probability. Vila Valério, ES–Brazil.

**Figure 2 plants-12-01451-f002:**
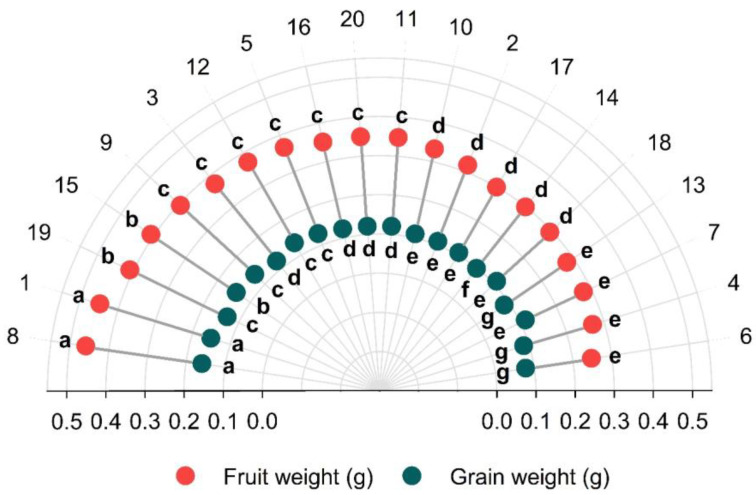
Weight of one fruit and one dry grain at 12% moisture of 20 *C. canephora* genotypes. Means followed by the same letter did not differ statistically by the Scott–Knott test at 5% probability. Vila Valério, ES–Brazil.

**Figure 3 plants-12-01451-f003:**
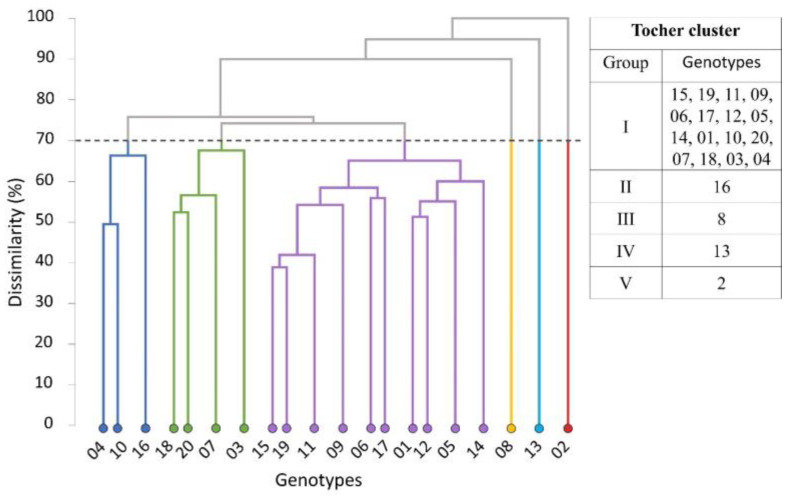
Grouping of 20 *C. canephora* genotypes by the UPGMA dendrogram and the Tocher method based on the Euclidean distance, considering grain and husk nutrient concentrations and the grain/husk percentage. Group I: light blue; group II: green; group III: purple; group IV: orange; group V: light blue; group VI: red. Cophenetic correlation: 0.80. Vila Valério, ES–Brazil.

**Figure 4 plants-12-01451-f004:**
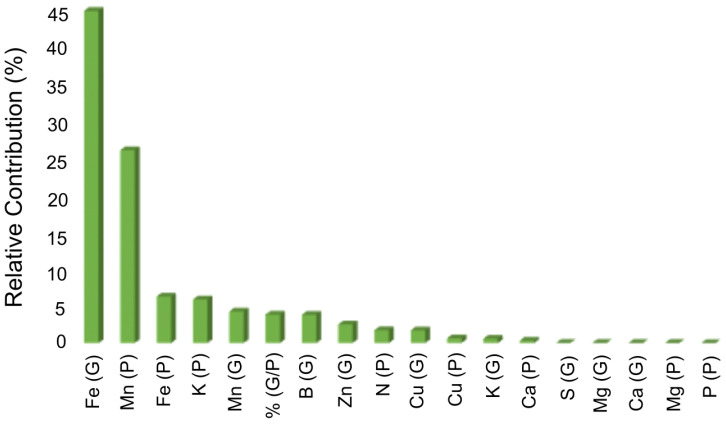
Relative contribution of grain and husk nutrient concentrations and grain/husk percentage in fruits for genetic diversity in 20 *C. canephora* genotypes using Singh’s method (1981). Vila Valério, ES–Brazil. G: grain; P: husk; Fe: iron; Mn: manganese; K: potassium; B: boron; Zn: zinc; N: nitrogen; Cu: copper; Ca: calcium; S: sulfur; Mg: magnesium; P: phosphorus.

**Figure 5 plants-12-01451-f005:**
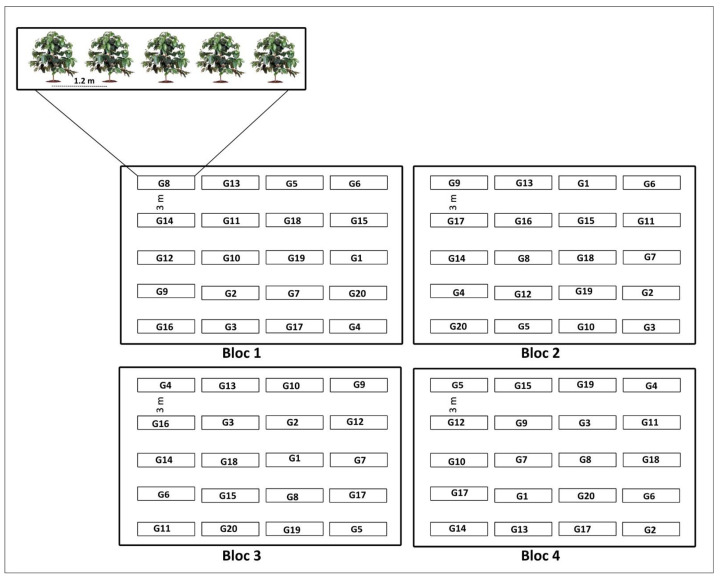
Experimental design of the area of 20 *C. canephora* genotypes. Vila Valério, State of Espírito Santo, Brazil. G: genotypes.

**Table 1 plants-12-01451-t001:** Analysis of variance and estimates of coefficient of experimental variation (CVe), coefficient of genetic variation (CVg) and heritability (H²) for nutrient concentrations in grain and husk and the percentage of grain and husk of 20 *C. canephora* genotypes. Vila Valério, State of Espírito Santo, Brazil.

Nutrients	Grain					Husk				
	*F*-Value	Mean	CVe (%)	CVg (%)	H² (%)	*F*-Value	Mean	CVe (%)	CVg (%)	H² (%)
N	1.89 ^ns^	29.28	5.47	2.98	47.15	11.64 **	16.67	4.99	9.41	91.41
P	1.63 ^ns^	1.79	9.75	4.45	38.51	6.45 **	1.10	4.5	6.06	84.50
K	2.76 *	17.01	7.17	3.83	46.09	5.37 **	24.99	9.02	10.89	81.39
Ca	7.95 **	1.65	7.11	10.82	87.42	11.33 **	4.61	8.03	14.90	91.17
Mg	6.42 **	1.68	8.47	11.39	84.43	15.82 **	0.89	8.57	19.05	93.68
S	2.03 *	1.65	16.15	9.47	50.79	1.47 ^ns^	1.54	20.67	8.21	32.14
Cu	10.26 **	9.42	9.37	16.45	90.25	10.30 **	6.27	8.70	15.32	90.29
Fe	6.80 **	19.12	28.92	40.21	85.30	2.40 *	22.68	15.35	10.50	58.42
Mn	11.61 **	19.40	6.74	12.67	91.38	35.29 **	16.71	11.07	37.43	97.17
Zn	2.62 **	9.10	23.17	17.01	61.8	0.80 ^ns^	10.06	18.53	0.00	0.00
B	4.28 **	10.37	19.64	20.53	76.62	1.35 ^ns^	21.02	18.11	6.18	25.88
	F Test (Gen)		Mean		CVe (%)		CVg (%)		H² (%)	
% Grain	9.78 **		56.95		2.74		4.06		89.77	
% Husk	9.78 **		43.05		3.62		4.06		89.77	

^ns^, ** and *, not significant, significant at 1 and 5% probability, respectively, by the F test. Gen: genotypes; N: nitrogen; P: phosphorus; K: potassium; Ca: calcium; Mg: magnesium; S: sulfur; Cu: copper; Fe: iron; Mn: manganese; Zn: zinc; B: boron.

**Table 2 plants-12-01451-t002:** Grain nutrient concentrations of 20 *C. canephora* genotypes. Vila Valério, Estate of Espírito Santo, Brazil.

Gen	Nutrients										
	k·g^−1^						mg·kg^−1^				
	N	P	K	Ca	Mg	S	Cu	Fe	Mn	Zn	B
1	29.17 a	1.83 a	16.70 b	1.60 b	1.57 c	1.40 b	8.73 d	13.53 c	19.00 b	6.73 b	9.10 b
2	28.00 a	1.77 a	17.97 a	1.90 a	2.23 a	1.70 b	6.87 e	42.33 a	23.43 a	10.50 a	9.63 b
3	28.93 a	1.90 a	17.17 b	1.67 b	1.73 c	1.97 a	12.17 a	29.20 b	23.40 a	11.65 a	14.60 a
4	32.43 a	2.03 a	16.23 b	2.00 a	1.90 b	1.63 b	10.90 b	13.70 c	18.77 b	8.60 b	8.97 b
5	28.93 a	1.77 a	16.27 b	1.53 b	1.53 c	1.47 b	9.50 c	12.43 c	17.13 c	5.75 b	12.07 a
6	30.10 a	1.77 a	16.70 b	1.47 b	1.57 c	1.80 a	9.67 c	12.60 c	19.50 b	7.53 b	12.83 a
7	28.93 a	1.83 a	18.57 a	1.90 a	1.57 c	1.40 b	9.80 c	18.55 c	17.00 c	10.93 a	8.87 b
8	27.30 a	1.70 a	17.83 a	1.40 b	1.40 c	1.53 b	8.50 d	14.60 c	14.87 c	10.40 a	7.93 b
9	28.00 a	1.63 a	17.53 a	1.50 b	1.77 b	1.57 b	9.37 c	16.40 c	18.60 b	11.40 a	11.30 a
10	31.03 a	1.87 a	16.40 b	1.80 a	1.70 c	1.73 b	9.63 c	12.90 c	23.33 a	7.63 b	5.60 b
11	29.40 a	1.60 a	17.17 b	1.47 b	1.60 c	2.10 a	9.53 c	18.40 c	18.57 b	8.40 b	8.40 b
12	29.63 a	1.53 a	16.00 b	1.43 b	1.37 c	1.57 b	7.03 e	13.40 c	17.37 c	8.47 b	9.35 b
13	30.57 a	1.73 a	16.83 b	1.97 a	2.03 a	1.40 b	12.50 a	32.07 b	18.30 b	12.13 a	15.67 a
14	30.10 a	1.83 a	15.93 b	1.60 b	1.80 b	1.63 b	8.40 d	17.53 c	22.90 a	8.80 b	10.33 b
15	29.87 a	1.87 a	17.93 a	1.60 b	1.67c	1.90 a	8.77 d	14.60 c	17.53 c	8.63 b	13.07 a
16	29.40 a	1.87 a	15.37 b	1.57 b	1.60 c	1.50 b	11.17 b	12.13 c	23.53 a	7.90 b	9.50 b
17	28.00 a	1.77 a	16.17 b	1.77 a	1.63 c	1.63 b	8.63 d	17.17 c	18.90 b	6.33 b	9.80 b
18	27.07 a	1.87 a	18.43 a	1.83 a	1.87 b	1.40 b	10.30 c	30.77 b	19.37 b	12.37 a	11.20 a
19	29.63 a	2.03 a	16.47 b	1.53 b	1.57 c	2.07 a	6.27 e	15.13 c	16.93 c	7.70 b	11.30 a
20	29.17 a	1.67 a	18.63 a	1.47 b	1.53 c	1.57 b	10.73 b	24.87 b	19.50 b	10.13 a	7.80 b

Means followed by the same letter in a column did not differ from each other by the Scott–Knott test at 5% probability. Gen: genotypes; N: nitrogen; P: phosphorus; K: potassium; Ca: calcium; Mg: magnesium; S: sulfur; Cu: copper; Fe: iron; Mn: manganese; Zn: zinc; B: boron.

**Table 3 plants-12-01451-t003:** Nutrient concentrations in the husk of 20 *C. canephora* genotypes. Vila Valério, ES–Brazil.

Gen	Nutrients										
	g·kg^−1^						mg·kg^−1^				
	N	P	K	Ca	Mg	S	Cu	Fe	Mn	Zn	B
1	15.63 b	1.10 b	27.50 a	6.10 a	0.87 d	1.53 a	6.67 c	28.53 a	19.30 c	9.40 a	23.23 a
2	18.90 a	1.07 b	21.43 b	4.47 b	1.13 b	1.97 a	4.07 e	24.80 b	35.03 a	9.97 a	19.33 a
3	18.20 a	1.10 b	26.20 a	4.53 b	1.00 c	1.60 a	6.53 c	20.85 b	16.87 d	9.63 a	24.27 a
4	16.80 a	1.07 b	26.53 a	6.30 a	1.23 a	1.37 a	7.13 b	22.10 b	22.67 b	11.70 a	22.77 a
5	15.63 b	1.03 c	17.70 c	4.93 b	0.73 d	1.57 a	6.13 c	22.27 b	18.20 c	10.63 a	23.63 a
6	17.27 a	1.23 a	29.40 a	4.07 c	0.90 c	1.90 a	6.93 b	19.95 b	13.77 d	9.47 a	19.20 a
7	15.63 b	1.03 c	25.27 a	4.63 b	0.70 e	1.47 a	5.80 d	22.07 b	11.30 e	9.27 a	20.30 a
8	14.70 c	1.07 b	18.93 c	3.60 c	0.53 f	1.47 a	5.00 d	32.23 a	8.90 e	9.20 a	19.50 a
9	16.10 b	1.07 b	22.53 b	3.53 c	0.67 e	1.30 a	5.40 d	23.75 b	14.97 d	9.03 a	19.57 a
10	17.97 a	1.03 c	27.83 a	5.13 b	1.07 b	1.23 a	6.87 b	21.00 b	17.77 c	11.07 a	24.20 a
11	14.23 c	1.10 b	23.93 a	4.37 b	0.77 d	1.50 a	6.33 c	19.80 b	11.93 e	9.23 a	17.50 a
12	16.57 b	1.00 c	26.37 a	5.00 b	0.93 c	1.37 a	6.17 c	20.90 b	20.03 c	9.17 a	17.53 a
13	18.43 a	1.27 a	25.50 a	4.27 b	1.10 b	1.67 a	7.40 b	21.80 b	13.53 d	10.17 a	23.30 a
14	14.00 c	1.00 c	25.67 a	4.93 b	0.83 d	1.37 a	5.70 d	19.40 b	23.70 b	9.60 a	21.23 a
15	15.17 c	1.17 a	24.17 a	4.30 b	0.77 d	1.25 a	5.47 d	24.55 b	9.60 e	9.23 a	24.17 a
16	19.40 a	1.20 a	29.00 a	4.63 b	1.03 b	1.43 a	8.63 a	21.30 b	18.57 c	10.40 a	20.43 a
17	17.50 a	1.13 b	23.63 a	3.47 c	0.93 c	1.90 a	6.17 c	19.97 b	10.93 e	12.20 a	15.70 a
18	17.97 a	1.10 b	26.60 a	4.73 b	0.90 c	1.47 a	6.53 c	21.60 b	14.57 d	9.77 a	23.30 a
19	14.93 c	1.10 b	26.23 a	4.50 b	0.80 d	1.57 a	5.13 d	22.63 b	9.30 e	10.23 a	19.33 a
20	18.43 a	1.13 b	25.30 a	4.67 b	0.80 d	1.90 a	7.23 b	24.10 b	23.17 b	11.80 a	21.97 a

Means followed by the same letter in a column did not differ from each other by the Scott–Knott test at 5% probability. Gen: genotypes; N: nitrogen; P: phosphorus; K: potassium; Ca: calcium; Mg: magnesium; S: sulfur; Cu: copper; Fe: iron; Mn: manganese; Zn: zinc; B: boron.

**Table 4 plants-12-01451-t004:** Nutrient accumulation in the fruit of 20 *C. canephora* genotypes in a ton of processed grain at 12% moisture. Vila Valério, ES–Brazil.

Gen	Nutrients										
	kg·ton^−1^						g·ton^−1^				
	N	P	K	Ca	Mg	S	Cu	Fe	Mn	Zn	B
1	37.48 e	2.44 c	34.46 b	5.73 b	2.03 f	2.35 g	12.61 e	32.31 f	30.80 e	12.71 g	24.56 d
2	38.72 d	2.36 d	31.37 c	4.86 d	2.83 a	2.92 b	9.10 i	55.89 a	45.89 a	16.54 c	22.33 e
3	38.03 d	2.44 c	32.49 c	4.44 e	2.22 e	2.82 c	15.27 b	39.98 d	32.14 d	16.80 c	28.91 b
4	42.15 b	2.65 a	34.75 b	6.57 a	2.66 b	2.51 e	15.29 b	29.11 g	34.15 c	16.64 c	25.31 d
5	36.92 e	2.31 d	26.81 e	4.74 d	1.89 h	2.40 f	12.80 e	26.42 i	27.93 f	12.45 g	27.02 c
6	40.73 c	2.56 b	37.94 a	4.48 e	2.12 f	3.11 a	14.16 c	26.90 i	28.39 f	14.18 e	26.53 c
7	35.36 f	2.27 e	31.51 c	4.41 e	1.83 i	2.12 i	12.27 f	29.64 g	22.00 i	15.31 d	19.82 f
8	33.46 g	2.17 f	27.34 e	3.39 g	1.59 j	2.26 h	10.67 h	32.22 f	18.75 j	14.86 d	18.68 f
9	36.95 e	2.25 e	32.03 c	3.89 f	2.08 f	2.35 g	12.37 f	31.87 f	27.60 f	16.82 c	24.25 d
10	40.48 c	2.40 c	33.92 b	5.14 c	2.27 d	2.42 f	13.45 d	26.07 i	33.35 c	14.50 e	21.62 e
11	36.22 f	2.19 f	31.56 c	4.24 e	1.97 g	2.91 b	12.89 e	29.93 g	24.83 g	13.79 f	19.30 f
12	37.92 d	2.06 g	32.03 c	4.62 d	1.86 i	2.33 g	10.48 h	26.04 i	29.07 f	13.78 f	20.13 f
13	43.37 a	2.65 a	36.86 a	5.40 c	2.78 a	2.68 d	17.62 a	47.59 b	28.08 f	19.63 a	33.94 a
14	38.20 d	2.44 c	34.36 b	5.27 c	2.28 d	2.54 e	12.05 f	30.96 f	39.17 b	15.43 d	25.83 c
15	38.59 d	2.58 b	34.53 b	4.70 d	2.09 f	2.67 d	12.09 f	31.79 f	23.18 h	14.82 d	30.11 b
16	38.53 d	2.43 c	31.62 c	4.25 e	2.09 f	2.24 h	15.41 b	23.99 j	32.69 d	13.51 f	21.07 e
17	36.29 f	2.31 d	29.32 d	3.75 f	2.06 f	2.66 d	11.67 g	27.96 h	23.95 g	13.31 f	18.63 f
18	37.56 e	2.49 c	35.93 a	5.08 c	2.35 c	2.33 g	14.07 c	43.54 c	28.11 f	18.28 b	26.97 c
19	37.58 e	2.63 a	33.67 b	4.60 d	1.99 g	3.00 b	9.36 i	29.89 g	22.02 i	14.29 e	24.05 d
20	38.26 d	2.24 e	33.06 b	4.31 e	1.90 h	2.63 d	14.37 c	37.96 e	32.49 d	16.73 c	21.08 e
Summary of analysis of variance											
Genotype	37.46 **	50.83 **	26.87 **	59.38 **	287.35 **	75.89 **	216.56 **	280.72 **	286.48 **	77.16 **	90.81 **
Mean	38.14	2.39	32.78	4.69	2.14	2.56	12.90	33.00	29.23	15.22	24.01
Cv %	1.94	1.99	3.34	3.97	1.75	2.50	2.25	2.94	2.55	2.82	3.63

Means followed by the same letter in a column did not differ from each other by the Scott–Knott test at 5% probability. Gen: genotypes; N: nitrogen; P: phosphorus; K: potassium; Ca: calcium; Mg: magnesium; S: sulfur; Cu: copper; Fe: iron; Mn: manganese; Zn: zinc; B: boron; ** significant at 5%.

**Table 5 plants-12-01451-t005:** Soil chemical and physical properties and particle-size fractions (0–20 cm layer) in the experimental area. Vila Valério, Espírito Santo, Brazil.

Chemical Characteristics												
mg·dm^−3^									cmol_c_·dm^−3^			
P	K	Na	S	B	Zn	Mn	Cu	Fe	Ca	Mg	Al	H + Al
58.5	86.0	3.0	18.0	0.4	7.4	9.1	0.9	48.0	1.8	0.3	0.2	4.5
pH			V (%)			S.B (cmol_c_·dm^−3^)			OM (dag·dm^−3^)			
5.0			34.0			2.3			2.0			
Particle-size fractions (g·kg^−1^)									Textural classification			
Sand			Silt			Clay			Clay sandy loam			
602			126			272						

pH: in H_2_O 1:2.5; P, K, Zn, Mn, Cu and Fe (phosphorus; potassium; zinc; manganese; copper, iron) Extraction: Mehlich^−1^; S (sulphur): Monocalcium phosphate acetic acid; Ca and Mg (calcium; magnesium) Extraction: 1 mol/L^−1^ KCI; H + Al (Hydrogen, aluminum) Titration; V (base saturation); SB (sum by bases); OM (organic matter): Embrapa method.

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
