# Peer review of "Nutritional Balance and Genetic Diversity of Coffea canephora Genotypes"

_plants, 2023, doi:10.3390/plants12071451_

Round 1

Reviewer 1 Report

Dear author(s),

there are some inspiring insights thorough the manuscript and I tend to agree on its publication. However, there are few points that needs to be quickly addressed to improve its overall communication:

Title:

1/ shortening advisable

Abstract:

2/ please understand that the purpose of the Abstract is to explain to all our readers (including those from other disciplines) what the paper is about, note that data such as "date of collection" are irrelevant

3/ do you really want to indicate that fertiliser recommendations can be made based on fruit properties?

Introduction:

4/ remove all clusters of references to avoid reference overkill (prefer only 1 reference to support 1 claim)

5/ complexity of phosphorus availability to organisms should be better explained, refer to Fig. 1 in paper "Novel sorbent shows promising financial results on P recovery from sludge water"

6/ make sure that this chapter fully introduces any reader into to the topic, explain all the terms, units, abbreviations, Latin and Greek letters, and the whole context that is necessary for anyone (including experts from other disciplines) to understand the following chapters

7/ do not ignore (economic) reality, comment on the complexity of production costs and profit forecasting, refer to papers "Does the life cycle affect earnings management and bankruptcy?" and "Predicting future Brent oil price on global markets"

8/ the research hypothesis could be stated more clearly, condensate the research hypothesis into 1 short statement (or question) that will be subsequently confirmed or refuted, make sure the urgency and significance of the research hypothesis was justified in its environmental - economic nexus

Materials and Methods:

9/ the method must be presented in such a way that it can be reproduced anytime, by anyone, anywhere (do not create obstacles like referring to specific location etc.)

10/ please understand that the methodology must be described in a completely unambiguous way that does not allow for multiple interpretations (everyone who reads this chapter should get very precise instructions on how to repeat your procedure to achieve exactly the same results)

11/ provide cost breakdown or at least some simplified financial analysis if you are about to argue that this concept is realistic

Results:

12/ each Tab. and Fig. should be provided with caption that describes A/ what can be seen and B/ how is this relevant to the research hypothesis

13/ avoid data overkill, present only the most most industrially important results

Discussion:

14/ show more self-criticism to your work (can all the methods and results be fully trusted? what are the weaknesses of the methods used? where do the main measurement inaccuracies arise? what are the limitations from a commercial point of view? are the lessons learned transferable to other fields?)

15/ data interpretation is very limited unless accompanied by economic interpretation, comment to latest findings in production/manufacturing management (refer to papers "Sustainable Industry 4.0 Wireless Networks, Smart Factory Performance, and Cognitive Automation in Cyber-Physical System-based Manufacturing" and "Data-driven Machine Learning and Neural Network Algorithms in the Retailing Environment: Consumer Engagement, Experience, and Purchase Behaviors")

16/ compare your results in more depth with the existing literature, identify the main deviations and try to explain the mechanisms by which they may have been caused

17/ discuss nutrient pricing and propose some improvements and direction for future research (refer to papers "Economic considerations on nutrient utilization in wastewater management" and "Silica nanoparticles from coir pith synthesized by acidic sol-gel method improve germination economics")

18/ reveal the main driving mechanisms of your results, provide deeper synthesis and reveal some more original/significant findings

Conclusions:

19/ present only original and industrially significant revelations that have the potential to expand the horizon of human knowledge (higher level of generalization is mandatory)

resumé:

this manuscript needs MINOR REVISION to: A/ be better communicated; B/ better address the economic reality.

Author Response

Por favor, verifique o anexo.

Reviewer 2 Report

The manuscript ‘Assessment of nutrient accumulation in fruit to enhance fertilizer management and recommendation for Coffea canephora’ needs major revision.

Add the novelty of the work in the last paragraph of the introduction.

Add the experimental design.

Discussion must be improved.

The conclusion must be rewritten. This seems like highlight.

Reviewer 3 Report

The topic of the study is of great scientific interest.
The introduction is satisfactory and substantiates the purpose of the research.
The proposed scientific hypothesis is confirmed.
The design and conduct of the experiment are justified, but explanations are needed
The authors are recommended to pay attention to the following:
1.    It is not clear what method was used to analyze the content of macro- and microelements in the studied samples; the method should be indicated.
2) It is more correct to evaluate the nutrient export taking into account the content of mineral elements also in the leaves.
Clarification of Figure 3 is required. The description of the genotypes in the text of the groups differs from that in the figure. Ъ
4 Does the sentence "Genotype 8 stands out for requiring a smaller amount of fruit to produce 1000kg of processed coffee" need clarification .....?
The authors used 55 literature sources related to the topic of the study.
Overall, the authors illustrated the article well with 6 tables and 4 figures, analyzed in detail the content of elements in coffee fruit, grouped the genotypes, and presented the results of cluster analysis.
The statistical analysis is compelling.
The article presents complete and interesting information on the mineral composition of coffee beans depending on the genotype.
However, I repeat, the studies conducted do not fully allow conclusions to be drawn about the mineral intake requirements of the different genotypes.
The article can be published in a journal after the shortcomings are eliminated.

Round 2

Reviewer 2 Report

This manuscript can be accepted in this present form.